# Effect of Moistube Fertigation on Infiltration and Distribution of Water-Fertilizer in Mixing Waste Biomass Soil

**Guangzhao Sun [1], Yilin Li [1], Xiaogang Liu [1,\*], Ningbo Cui [2], Yanli Gao [1] and Qiliang Yang [1]**

[1]  Faculty of Agriculture and Food, Kunming University of Science and Technology, Kunming 650500, China; sunguangzhao@stu.kust.edu.cn (G.S.); liyilin1@stu.edu.cn (Y.L.); gaoyanli@stu.kust.edu.cn (Y.G.); 20090042@kust.edu.cn (Q.Y.)

[2]  State Key Laboratory of Hydraulics and Mountain River Engineering and College of Water Resource and Hydropower, Sichuan University, Chengdu 610065, China; cuiningbo@scu.edu.cn

\*  Correspondence: liuxiaogangjy@126.com or liuxiaogangjy@kust.edu.cn

**Abstract:** A series of indoor soil box simulation experiments were carried out to investigate the infiltration capacity of fertilizer solution in mixing waste biomass and the distribution characteristics of water-fertilizer in wetted soil under moistube fertigation. The infiltration rate and cumulative infiltration of moistube fertigation in soils as well as the distribution characteristics of water-fertilizer (soil water, nitrate–N, available P, and available K) in wetted soil were studied in three waste biomass (peanut shell) mixing ratios (MR1.5%, MR3.0%, and MR4.5%) taking a not amended soil as control (CK). The cumulative infiltration of fertilizer solution and the distribution of water-fertilizer were fitted by a modified infiltration model. Results indicated that increasing the mixing ratio improved significantly the infiltration rate and cumulative infiltration of fertilizer solution and the distribution area and content of water-fertilizer in amended wetting soil compared with CK. The relationship between the cumulative infiltration of fertilizer solution and infiltration time conformed to the Kostiakov infiltration model. The distribution uniformity coefficient of soil water and nitrate–N increased with the increase in waste biomass mixing ratio, whereas available P and available K decreased in wetted soil. The 4-parameter log-logistic model fitted well with the distribution of water-fertilizer in mixing waste biomass wetted soil under moistube fertigation. The research results could provide a theoretical basis and practical reference for the popularization and application of new moistube fertigation technology.

**Keywords:** irrigation and fertilization; waste biomass; infiltration characteristics; nutrient distribution

## 1. Introduction

The average annual total of agricultural irrigation water is very high in China, but the water use efficiency is relatively low [1]. Similarly, China's fertilizer use is high per year, but the utilization rate of fertilizer is low in the current season [2]. Moreover, the excessive use of fertilizer increases the agricultural cost and causes environmental pollution. Research reported that there are many questions about cultivated land in China. For example, the area of inferior cultivated land was high, and the soil organic matter was decreased [3]. Thus, it is greatly significant to employ both effective fertigation method and agricultural waste biomass returning field technology in China for improving the utilization efficiency of agricultural resources and for achieving the sustainable development of agriculture.

Fertigation is a new technique that combines irrigation with fertilization to fertilize through irrigation systems. This technology has the characteristics of fast fertilizer efficiency, saving water

and fertilizer, and reducing production cost [4]. At present, drip and sprinkler fertigation technology are widely used [5–7] and there are many research results on the infiltration characteristics and distribution of soil water-fertilizer under drip fertigation [8–11]. Numerous studies revealed that the movement and distribution of nitrogen in soil were affected by various factors, such as soil properties, drip flow, fertilizer solution concentration, fertilizer type, fertilization time, and frequency under drip fertigation [12–16]. Suitable drip outflow velocity and fertilization frequency reduced nitrogen leaching in soil and improved nitrogen fertilizer utilization efficiency [17,18].

Moistube irrigation is a new underground precision micro-irrigation technology based on micro-nano porous membrane. Moistube irrigation is driven by the water potential gradient inside and outside of membrane and soil suction. It delivers irrigation water directly into the crop root zone continuously and slowly [19,20], thus providing an effective carrier for agricultural fertigation technology. It has the advantages of reducing ground evaporation and of improving irrigation water use efficiency [19–22]. Several studies have shown that the soil physical properties (texture, bulk density, and initial water content), the emitter parameters (outflow velocity, length, and buried depth), the pressure head, and the salinity of irrigation water mainly influenced the infiltration and distribution of soil water under moistube irrigation [20,23–27]. The wetted soil is approximately a symmetrical ellipsoid with moistube as the axis [20]. Soil water content reduced gradually further from the source to all around, and the maximum value appeared around the moistube [20,28,29]. Currently, the infiltration characteristics of fertilizer solution and the distribution of water-fertilizer in wetted soils during moistube fertigation are not known.

Agricultural waste biomass returning technology is an effective method to improve the ecological environment of farmland and to promote the sustainable development of agriculture. It can effectively ameliorate soil physical structure, enhance soil infiltration and water retention capacity, and improve soil organic matter content and nutrient element reserves [30–34]. Previous studies have shown that comminuted straw amendment rapidly improves soil structure and increases the steady infiltration rate and cumulative infiltration of soil, while mixing the long straw decreases the soil infiltration capacity [35,36].

Peanut is a widely cultivated edible nut in the world [37]. The worldwide peanut planting area, yield, and shell waste were about 21 million hectares, 200 million tons, and 6 million tons, respectively, in 2010 [38,39]. Peanut shell returning will certainly improve soil physicochemical properties and soil fertility; however, how mixing waste biomass (peanut shell) in the soil influences the infiltration characteristics of fertilizer solution and the distribution of water-fertilizer under moistube fertigation is still unsolved and needs further investigation.

In this study, we investigated that the infiltration capacity of fertilizer solution in mixing waste biomass soil and the distribution of water-fertilizer in wetted soil under moistube fertigation. The cumulative infiltration of fertilizer solution and the distribution of water-fertilizer in wetted soil were fitted according to the model. The objective was to provide a theoretical basis and practical reference for the popularization and application of new moistube fertigation technology. It is significant to apply novel irrigation strategy and agricultural waste biomass returning field in the recycling of waste biomass resources, especially in water resource-scarce regions.

## 2. Materials and Methods

### 2.1. Experimental Materials

The experimental soil was taken from the superficial soil layer (0–20 cm) of the agricultural water-saving experimental field of Kunming University of Science and Technology (24°9′ N, 102°79′ E; 1978.9 m above sea level). The soil was air-dried and sieved in order to pass through a 2-mm sieve for reserve. Soil particle size distribution was analyzed using laser particle size analyzer (Mastersizer-2000, Malvern Panalytical, Malvern, UK), and soil type was determined to be clay loam, according to the

international soil classification system [19]. Table 1 shows the basic physicochemical properties of experimental soil.

Sieved soil was mixed uniformly with peanut shell by the design ratio. The experimental fertilizer was water-soluble fertilizers (Stanley Agriculture Group Co., Ltd., Linyi, China), and the contents of N, $P_2O_5$, and $K_2O$ were all 20%. The concentration of infiltration fertilizer solution was designed for 400 mg $L^{-1}$.

**Table 1.** Basic physicochemical properties of experimental soil.

| Soil Properties | Values |
|---|---|
| Soil mechanical composition (%) | |
| Sand (0.05–2 mm) | 16.8 |
| Silt (0.002–0.05 mm) | 28.63 |
| Clay (0–0.002 mm) | 54.57 |
| Basic hydraulic parameters (%) | |
| Initial soil water content | 3.61 |
| Saturated soil water content | 47.52 |
| Soil nutrient content (mg $kg^{-1}$) | |
| Nitrate–N | 26.75 |
| Available P | 4.42 |
| Available K | 35.86 |

## 2.2. Experimental Device

The experimental device consisted of five parts, i.e., the soil box, the height adjustable stand, the Mariotte bottle, the hydraulic hose, and the moistube (Figure 1). The soil box (40 cm × 40 cm × 45 cm) was made of transparent acrylic with the thickness of 10 mm. The Mariotte bottle (8 cm in diameter and 60 cm in height) provided a constant pressure head. The 3rd generation moistube (Moistube Irrigation Co., Ltd., Shenzhen, China) was a four-fold double-layer structure, in which the microporous inner layer (with a density of micropores higher than $10^5$ per $cm^2$ and a diameter ranging from 10 nm to 900 nm) was a micro-nano porous membrane with thickness of 0.06 mm, and the outer layer was a non-woven protective layer. The moistube had a 16-mm inside diameter, a folding diameter (width) of 25 ± 1.5 mm, and a thickness of 0.9 ± 0.5 mm. The moistube was arranged in a vertical inserting mode. The top of the moistube was connected to the Mariotte bottle through a joint and hydraulic hose, and the bottom was closed with a rubber plug.

## 2.3. Experimental Design and Method

Three waste biomass (peanut shell) mixing mass ratios were included in the experiment, and they were 1.5% (MR1.5%), 3.0% (MR3.0%), and 4.5% (MR4.5%). No mixing peanut shell (MR0%) was taken as the control (CK). This experiment consisted of 4 treatments, and each treatment was replicated 3 times. All the data in this paper are reported as means.

The experimental soil with the designed bulk density of 1.20 g·cm$^{-3}$ was layered (5 cm per layer) into the soil box, and loading depth was 40 cm. The effective infiltration length of the moistube was 30 cm, and the upper joint was flush with the soil surface. The pressure head was 1.0 m in the experiment.

The fertilizer solution level of the Mariotte bottle was recorded immediately after the moistube was filled with fertilizer solution and buried in the soil. The experimental data were recorded once every hour in 0–10 h, once every 2 h in 10–24 h, once every 4 h in 24–60 h, and once every 8 h in 60–124 h. Finally, after infiltration for 124 h, the fertilizer solution supply was stopped and soil samples in wetted soil were collected instantly by the layered sampling method, with a distance between sampling points

of 5 cm. The first point of each layer was taken around the moistube, and one layer was taken every 5 cm in the vertical direction. The distribution of sampling points is shown in Figure 2.

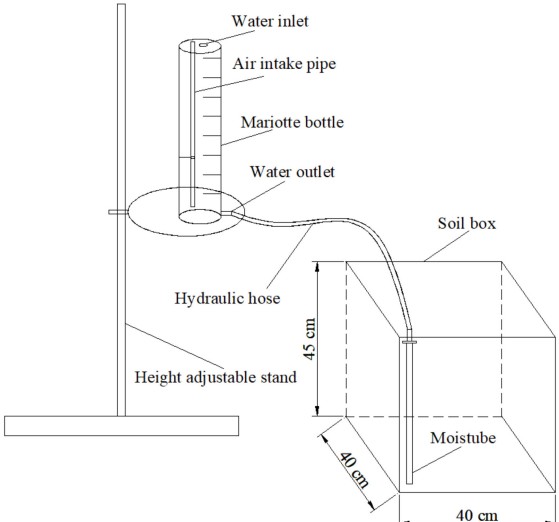

**Figure 1.** Schematic diagram of the experimental device.

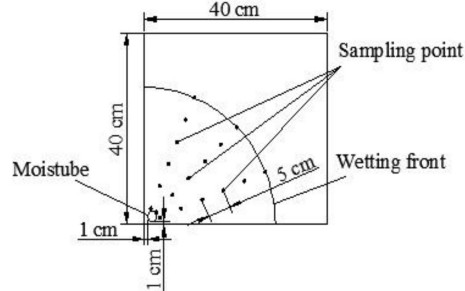

**Figure 2.** Distribution diagram of sample points.

## 2.4. Water-Fertilizer Content Measurement

Soil water content was calculated using oven-drying method. The measured soil nutrient is based on the method used by Wang R.S. et al. [8]. Nitrate–N was extracted using 1 mol $L^{-1}$ KCl and measured using UV spectrophotometer. Available P was extracted using 0.5 mol $L^{-1}$ $NaHCO_3$ and measured using Mo–Sb colorimetry method. Available K was measured using flame photometer method.

## 2.5. Models Applied

The whole process of fertilizer solution infiltration was divided into three stages. We took the average infiltration rate of 0–24 h as the average infiltration rate of first stage (AIRSI); likewise, the average infiltration rate of second stage (AIRSII) and third stage (AIRSIII) were for the periods 24–60 h and 60–124 h, respectively. The average infiltration rate of the first 6 h was taken as the initial infiltration rate (IIR), the average infiltration rate of the final 24 h was taken as the steady infiltration rate (SIR), and the average infiltration rate of the 0–124 h period was taken as the overall average infiltration rate (AIR).

Kostialov infiltration model was expressed in Equation (1) as follows:

$$I_t = Kt^{\alpha} \tag{1}$$

where $I_t$ is the cumulative infiltration (L), $K$ is infiltration coefficient (L $h^{-1}$), $\alpha$ is infiltration index, and $t$ is infiltration time (h).

Christensen's uniformity coefficient was calculated by Equation (2) as follows:

$$C_u = \left(1 - \frac{\sum_{i=1}^{m} |\theta_i - \overline{\theta}|}{m \times \overline{\theta}}\right) \times 100\% \tag{2}$$

where $C_u$ is the distribution uniform coefficient of water-fertilizer (%), $\overline{\theta}$ is the average content of water-fertilizer (% or mg·kg$^{-1}$), $\theta_i$ is the water-fertilizer content of the $i$th soil sample (% or mg kg$^{-1}$), and $m$ is the number of sampling points.

The 4-parameter log-logistic model was expressed in Equation (3) as follows:

$$C = A + \frac{B - A}{1 + 10^{k(l - lgD)}} \ (0 \leq l \leq 25) \tag{3}$$

where $C$ is the fitted value of water-fertilizer content (% or mg·kg$^{-1}$), $A$ is the lower limit of water-fertilizer content (% or mg·kg$^{-1}$), $B$ is the upper limit of water-fertilizer content (% or mg·kg$^{-1}$), lg$D$ is the horizontal distance when the decreasing rate of water-fertilizer content begins to change (cm), $k$ is the decreasing rate parameter of water-fertilizer content, and $l$ is the horizontal distance (cm).

### 2.6. Statistical Analysis

The experimental data were analyzed using Microsoft Excel 2010. Variance analysis (ANOVA) was performed using Statistical Product and Service Solutions (SPSS) 21.0 (International Business Machines Corporation, Armonk, NY, USA). The distribution graphics of water-fertilizer were drawn using Surfer 11.0 (Golden Software, Golden, CO, USA), and the water-fertilizer distribution area of wetting pattern profile was calculated by programming with MATLAB 7.0 (Math Works, Natick, MA, USA).

The accuracy of the fitted model was evaluated using root mean square error (*MAE*), mean relative error (*RMSE*), and normalized root mean squared error (*NRMSE*). These parameters are defined by the Equations (4)–(6) below:

$$MAE = \frac{1}{n} \sum_{i=1}^{n} \left| Y_i^{mea} - Y_i^{fit} \right| \tag{4}$$

$$RMSE = \sqrt{\frac{1}{n} \sum_{i=1}^{n} \left( Y_i^{mea} - Y_i^{fit} \right)^2} \tag{5}$$

$$NRMSE = \frac{\sqrt{\frac{1}{n} \sum_{i=1}^{n} \left( Y_i^{mea} - Y_i^{fit} \right)^2}}{Y_M} \times 100\% \tag{6}$$

where $Y_i^{mea}$ is the measured value, $Y_i^{fit}$ is the fitted value, $Y_M$ is the average of the measured value, and $n$ is the total number of measurements.

## 3. Results

### 3.1. The Infiltration Characteristics of Fertilizer Solution in Mixing Waste Biomass Soil Under Moistube Fertigation

#### 3.1.1. The Infiltration Rate of Fertilizer Solution

Under moistube fertigation, the IIR and AIRSI of fertilizer solution infiltration were higher while SIR and AIRSIII were lower (Table 2). Mixing waste biomass in soil significantly increased the infiltration rate of fertilizer solution ($p < 0.05$). Compared with CK, mixing waste biomass increased AIR, AIRSI, AIRSII, AIRSIII, IIR, and SIR by 50.4–96.6%, 54.1–108.4%, 37.9–126.0%, 52.7–104.6%, 22.8–61.5%, and 38.7–109.1%, respectively. The six infiltration rate parameters increased with the increase of waste biomass mixing ratio.

**Table 2.** Infiltration rate of fertilizer solution in mixing peanut shell soil under moistube fertigation.

| Treatments | Infiltration Rate (mL h$^{-1}$) | | | | | |
|---|---|---|---|---|---|---|
| | AIRSI | AIRSII | AIRSIII | IIR | SIR | AIR |
| MR0% | 27.20 ± 2.38 d | 17.45 ± 2.63 d | 14.37 ± 2.49 c | 45.78 ± 4.26 c | 14.25 ± 2.02 c | 22.84 ± 2.76 c |
| MR1.5% | 41.91 ± 3.11 c | 24.07 ± 2.65 c | 21.94 ± 1.26 b | 56.19 ± 4.46 b | 19.77 ± 1.84 b | 34.36 ± 2.29 b |
| MR3.0% | 48.95 ± 3.40 b | 30.81 ± 1.87 b | 23.47 ± 3.17 b | 66.61 ± 5.21 a | 24.50 ± 4.29 ab | 37.47 ± 4.62 ab |
| MR4.5% | 56.68 ± 2.02 a | 39.43 ± 2.56 a | 29.40 ± 3.03 a | 73.92 ± 3.93 a | 29.80 ± 2.62 a | 44.90 ± 4.02 a |

(1) MR means mixing ratios of waste biomass in soil; AIRSI, AIRSII, and AIRSIII indicate the average infiltration rate of the first stage (0–24 h), the second stage (24–60 h), and the third stage (60–124 h), respectively. (2) IIR SIR AIR means the initial infiltration rate (the average infiltration rate of the first 6 h), the steady infiltration rate (the average infiltration rate of the final 24 h), and the overall average infiltration rate (the average infiltration rate of the 0–124 h period). (3) Date are average values ± SD. (4) Different letters indicate statistically significant differences at $p < 0.05$.

### 3.1.2. The Infiltration Rate of Fertilizer Solution

As shown in Figure 3, the cumulative infiltration of fertilizer solution increased with the increase of waste biomass mixing ratio in the soil. Compared with CK, the cumulative infiltrations of MR1.5%, MR3.0%, and MR4.5% were improved by 49.2, 81.0, and 120.3%, respectively.

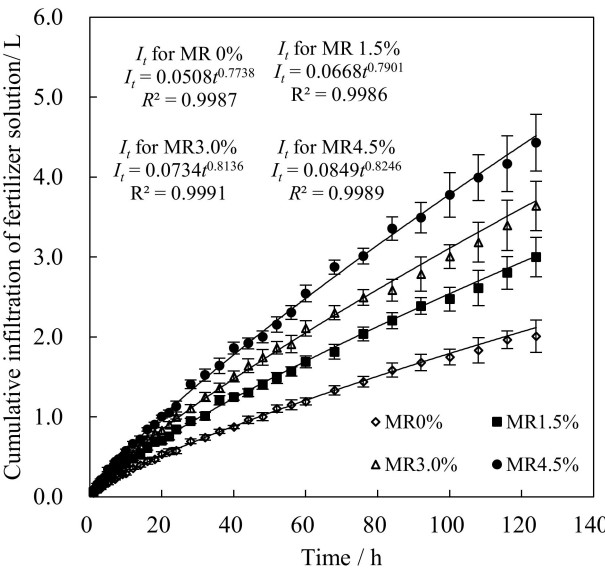

**Figure 3.** Cumulative infiltration of fertilizer solution in mixing waste biomass soil under moistube fertigation. MR means mixing ratios of peanut shell in soil; $I_t$ is the cumulative infiltration; $t$ is infiltration time.

### 3.1.3. The Infiltration Rate of Fertilizer Solution

The measured data of cumulative infiltration of fertilizer solution under different waste biomass mixing ratios were fitted with the Kostiakov infiltration model (Figure 3), and the determination coefficients $R^2$ were all greater than 0.99. In addition, it was found that the infiltration coefficient $K$ had a linear relationship with the waste biomass mixing ratio (MR) and that the infiltration index $\alpha$ had an exponential relationship with MR. The results of regression analysis were as follows:

$$K = 0.0073MR + 0.0526 \left( R^2 \ = \ 0.9751 \right) \tag{7}$$

$$\alpha = 0.7744e^{0.0145MR} \left( R^2 \ = \ 0.9798 \right) \tag{8}$$

The regression equation of infiltration parameters in Equation (7) and Equation (8) were substituted into Equation (1) to obtain the relationship model between the cumulative infiltration of fertilizer solution and waste biomass mixing ratio and infiltration time (Equation (9)):

$$I_t = (0.0073MR + 0.0526)t^{(0.7744e^{0.0145MR})} \tag{9}$$

Error analysis between the fitted values and the measured values were conducted to verify the reliability of the above model (Table 3). The results showed that the variation range of *MAE* and *RMSE* of the cumulative infiltration model both were small and that the *NRMSE* was less than 10%, indicating that the model had higher precision.

**Table 3.** Error analysis of the cumulative infiltration model of moistube fertigation under different peanut shell mixing ratios.

| Treatments | Model Error | | |
|---|---|---|---|
| | *MAE* (L) | *RMSE* (L) | *NRMSE* (%) |
| MR0% | 0.035 | 0.056 | 6.927 |
| MR1.5% | 0.045 | 0.055 | 4.877 |
| MR3.0% | 0.032 | 0.043 | 3.171 |
| MR4.5% | 0.045 | 0.062 | 3.778 |

(1) MR means mixing ratios of waste biomass in soil; (2) *MAE*, *RMSE*, and *NRMSE* represent the root mean square error, mean relative error, and normalized root mean squared error of the fitted model, respectively.

### 3.2. The Distribution of Water-Fertilizer in Mixing Waste Biomass Soil Under Moistube Fertigation

#### 3.2.1. The Distribution Characteristics of Water-Fertilizer

Under moistube fertigation, the contour of water-fertilizer content in soil profile was approximately ellipsoid around the moistube (Figure 4). The maximum content of water-fertilizer occurred just next to the moistube, the water-fertilizer content gradually decreased with the increase of horizontal distance, and the minimum value was close to the initial value in wetted soil. Compared with CK, mixing waste biomass significantly increased the maximum content of water-fertilizer in wetted soil from 24.0 to 78.4%.

The distribution of soil water and nitrate–N were similar, and the distribution range was larger in the soil profile. Meanwhile, the distribution of available P and available K tended to be the same and the distribution range were smaller and accumulated around the moistube (Figure 4).

The distribution area of water-fertilizer in wetted soil was significantly diverse under different waste biomass mixing ratios (Figure 4). Compared with CK, the MR1.5%, MR3.0%, and MR4.5% treatments improved the distribution area of soil water greater than 10% by 5.4, 8.8, and 19.6%; the distribution area of nitrate–N content was greater than 35 mg kg$^{-1}$ by 9.0, 14.4, and 26.5%; the distribution area of available P content was greater than 12 mg kg$^{-1}$ by 21.9, 26.4, and 45.7%; and the distribution area of available K content was greater than 100 mg kg$^{-1}$ by 5.5, 7.7, and 9.5%, respectively.

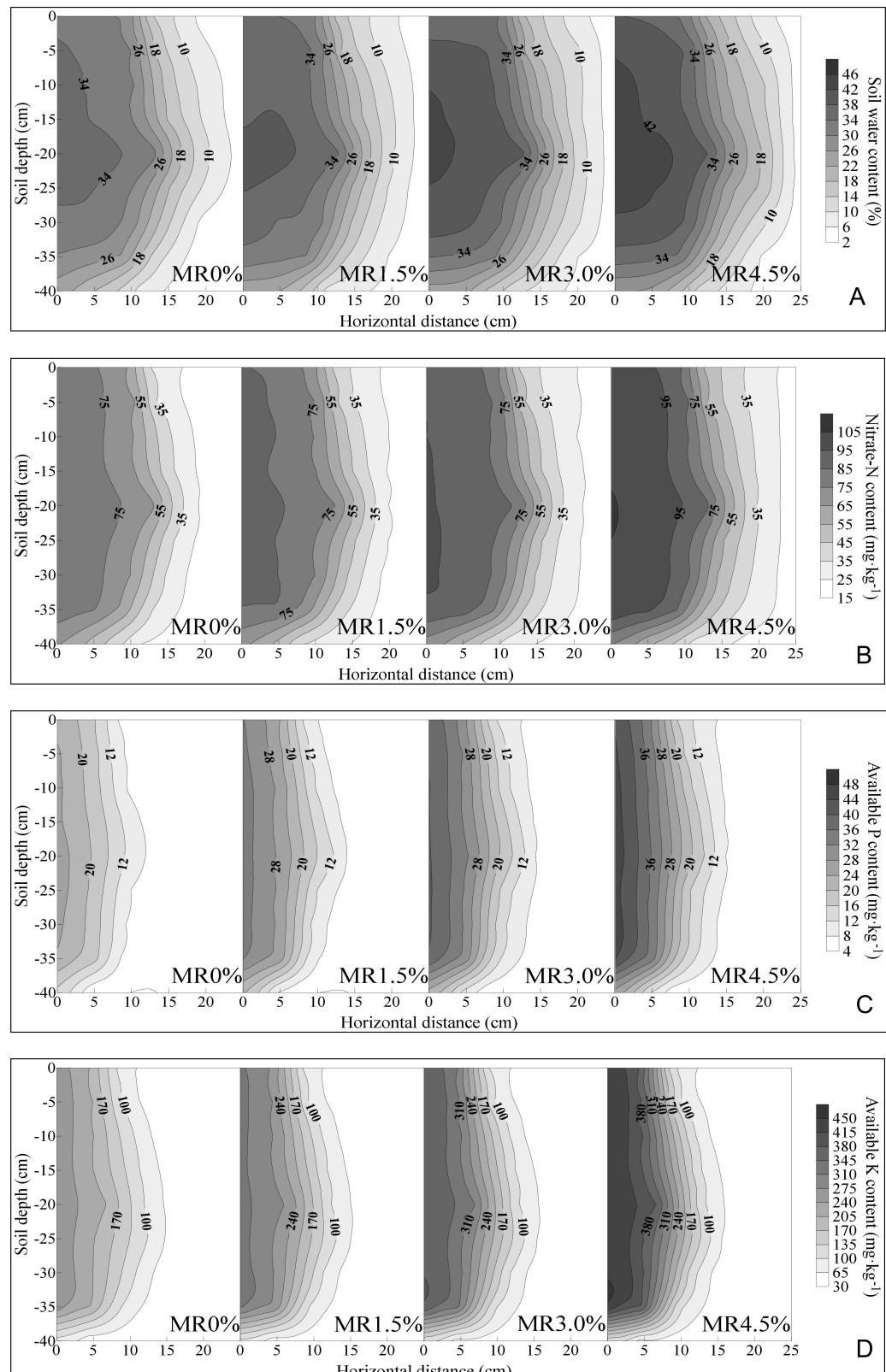

**Figure 4.** (**A–D**) The distribution of the soil water content, nitrate–N, available P, and available K, respectively, in wetted soil of moistube fertigation under different peanut shell mixing ratios. MR means mixing ratios of waste biomass in soil.

### 3.2.2. The Average Content and Distribution Uniformity of Water-Fertilizer

Statistical analysis (Table 4) showed that waste biomass mixing ratio had a significant effect on the average content of water-fertilizer in wetted soil of moistube fertigation ($p < 0.05$). Compared with CK, MR1.5%, MR3.0%, and MR4.5% increased the average content of soil water by 20.8, 34.4, and 44.7%; of nitrate–N by 10.9, 19.3, and 40.7%; of available P by 35.0, 74.2, and 122.3%; and of available K by 33.7, 56.4, and 95.8%, respectively. Thus, mixing waste biomass in the soil increased the content of water-fertilizer in wetted soil of moistube fertigation and the content of water-fertilizer increased as the increase of waste biomass mixing ratio.

In the experiment, the Christensen's uniformity coefficient [40] was used to quantitatively evaluate the distribution uniformity of water-fertilizer in wetted soil of moistube fertigation (Table 4). The distribution uniformity coefficients of soil water and nitrate–N were high, while that of available P and K were low. Moreover, there was a remarkable effect of the waste biomass mixing ratio on the distribution uniformity coefficient of water-fertilizer ($p < 0.05$). Compared with CK, the MR1.5%, MR3.0%, and MR4.5% treatments raised the distribution uniformity coefficient of soil water by 9.0, 15.1, and 21.0% and that of nitrate–N by 11.3, 17.7, and 25.1% while reduced that of available P by 6.3, 14.5, and 25.9% and that of available K by 13.4, 23.9, and 32.8%, respectively.

### 3.2.3. Model Fitting of Water-Fertilizer Distribution

The content of water-fertilizer in wetted soil of moistube fertigation showed an inverse S-shaped decreasing trend with the increase of horizontal distance. In this paper, the 4-parameter log-logistic model was used to fit the relationship between the content of water-fertilizer and horizontal distance in wetted soil under different waste biomass mixing ratios (Table 5). The results showed that the determination coefficients $R^2$ of the fitted equation were all greater than 0.95 and passed the significance test of $p < 0.05$.

Comparing the fitting values of water-fertilizer content with the measured values in wetted soil of moistube fertigation, it was found that the overall fitting effect was better (Figure 5). The content of soil water, nitrate–N, and available P and K in wetted soil decreased with the increase of horizontal distance and increased with the increase of waste biomass mixing ratio. In addition, the model error analysis results (Table 6) showed that the variation range of the *MAE*, *RMSE*, and *NRMSE* were all small. Therefore, the 4-parameter log-logistic model could fit well with the relationship between the content of water-fertilizer and horizontal distance in wetted soil of moistube fertigation under mixing waste biomass soil.

**Table 4.** Average content and the distribution uniformity coefficient of water-fertilizer in wetted soil of moistube fertigation under different peanut shell mixing ratios.

| Treatments | Soil Water Content | | Nitrate–N | | Available P | | Available K | |
|---|---|---|---|---|---|---|---|---|
| | Average (%) | $C_u$ (%) | Average (mg kg$^{-1}$) | $C_u$ (%) | Average (mg kg$^{-1}$) | $C_u$ (%) | Average (mg kg$^{-1}$) | $C_u$ (%) |
| MR0% | 25.6 ± 2.6 c | 74.2 ± 1.8 c | 69.6 ± 4.8 c | 68.5 ± 3.0 d | 14.7 ± 1.0 d | 63.0 ± 2.5 a | 132.8 ± 7.5 d | 61.1 ± 2.7 a |
| MR1.5% | 31.0 ± 2.7 b | 80.8 ± 4.2 bc | 77.1 ± 4.4 bc | 76.3 ± 2.3 c | 19.9 ± 1.5 c | 59.1 ± 3.4 ab | 177.6 ±14.1 c | 52.9 ± 2.6 b |
| MR3.0% | 34.5 ± 2.0 ab | 85.4 ± 3.4 ab | 83.0 ± 4.3 b | 80.7 ± 1.4 b | 25.7 ± 2.9 b | 53.9 ± 3.8 c | 207.7 ± 10.4 b | 46.4 ± 2.5 c |
| MR4.5% | 37.1 ± 1.9 a | 89.7 ± 4.5 a | 97.9 ± 4.4 a | 85.7 ± 2.1 a | 32.7 ± 2.3 a | 46.7 ± 1.6 d | 260.1 ± 13.0 a | 41.0 ± 2.0 d |

(1) MR means mixing ratios of waste biomass in soil; (2) $C_u$ indicates the Christensen's uniformity coefficient.

**Table 5.** Model fitting of water-fertilizer distribution in wetted soil of moistube fertigation under different peanut shell mixing ratios.

| Fitted Index | Fitted Result | Treatments | | | |
|---|---|---|---|---|---|
| | | MR0% | MR1.5% | MR3.0% | MR4.5% |
| Soil water content (%) | Fitted equation | $C = 3.40 + \dfrac{32.41 - 3.40}{1 + 10^{0.17(l-14.11)}}$ | $C = 3.52 + \dfrac{35.15 - 3.52}{1 + 10^{0.16(l-14.50)}}$ | $C = 3.13 + \dfrac{38.48 - 3.13}{1 + 10^{0.14(l-14.94)}}$ | $C = 3.04 + \dfrac{40.60 - 3..04}{1 + 10^{0.12(l-15.54)}}$ |
| | $R^2$ | 0.97 * | 0.95 * | 0.95 * | 0.99 * |
| Nitrate–N (mg kg$^{-1}$) | Fitted equation | $C = 18.99 + \dfrac{79.11 - 18.99}{1 + 10^{0.16(l-12.16)}}$ | $C = 18.95 + \dfrac{84.45 - 18.95}{1 + 10^{0.16(l-13.25)}}$ | $C = 19.80 + \dfrac{91.53 - 19.80}{1 + 10^{0.15(l-13.32)}}$ | $C = 18.74 + \dfrac{100.57 - 18.74}{1 + 10^{0.14(l-13.90)}}$ |
| | $R^2$ | 0.96 * | 0.97 * | 0.97 * | 0.98 * |
| Available P (mg kg$^{-1}$) | Fitted equation | $C = 4.41 + \dfrac{24.73 - 4.41}{1 + 10^{0.22(l-5.82)}}$ | $C = 4.30 + \dfrac{32.83 - 4.30}{1 + 10^{0.19(l-6.86)}}$ | $C = 4.33 + \dfrac{40.50 - 4.33}{1 + 10^{0.17(l-6.84)}}$ | $C = 4.46 + \dfrac{44.48 - 4.46}{1 + 10^{0.16(l-7.62)}}$ |
| | $R^2$ | 0.99 * | 0.99 * | 0.99 * | 0.98 * |
| Available K (mg kg$^{-1}$) | Fitted equation | $C = 34.87 + \dfrac{253.46 - 34.87}{1 + 10^{0.18(l-6.81)}}$ | $C = 37.78 + \dfrac{296.85 - 37.78}{1 + 10^{0.23(l-7.41)}}$ | $C = 37.99 + \dfrac{365.04 - 37.99}{1 + 10^{0.22(l-7.19)}}$ | $C = 39.42 + \dfrac{414.68 - 39.42}{1 + 10^{0.25(l-7.51)}}$ |
| | $R^2$ | 0.98 * | 0.99 * | 0.99 * | 0.99 * |

(1) MR means mixing ratios of waste biomass in soil; (2) $C$ means the fitted value of water-fertilizer content; (3) $l$ means the horizontal distance; (4) asterisks indicate significant correlations based on regression analyses (* $p < 0.05$).

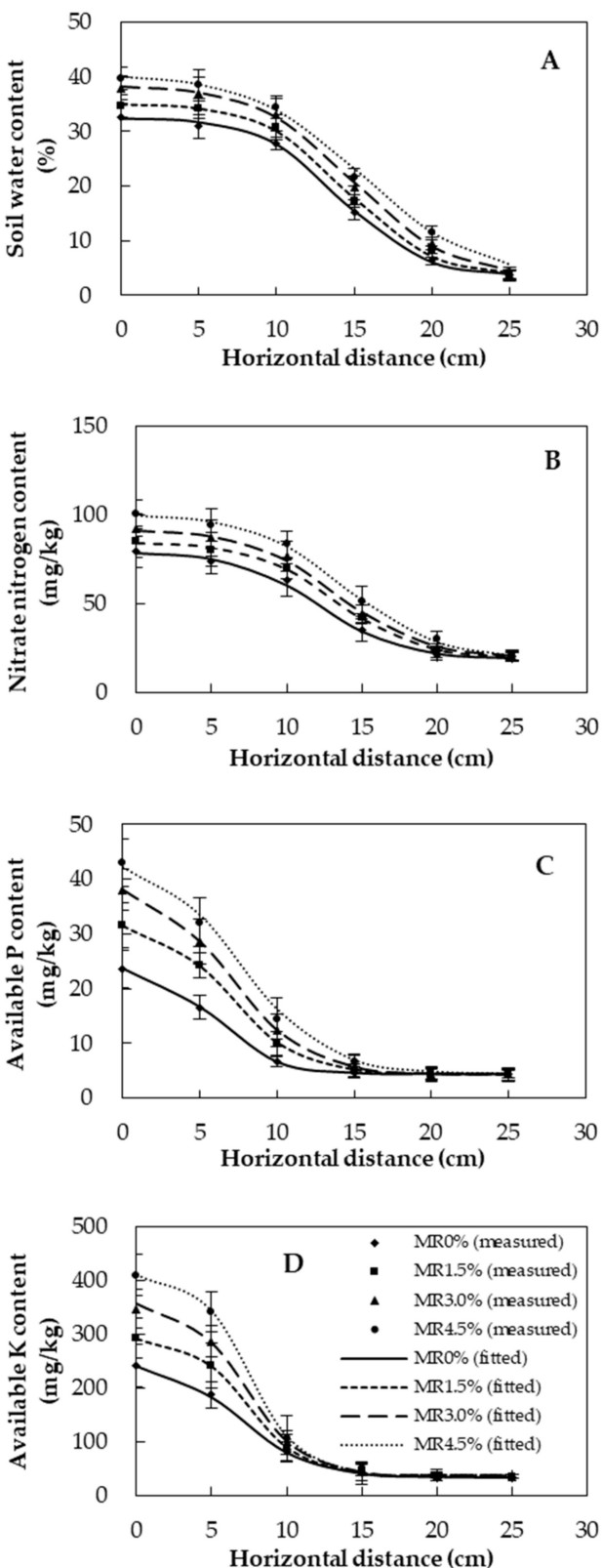

**Figure 5.** (**A–D**) Fitted and measured values of the soil water content, nitrate–N, available P, and available K, respectively, in wetted soil by moisture under different peanut shell mixing ratios. MR means mixing ratios of waste biomass in soil.

**Table 6.** Model error analysis of water-fertilizer distribution in wetted soil of moistube fertigation under different peanut shell mixing ratios.

| Index | Treatment | Model Error | | |
|---|---|---|---|---|
| | | MAE (% or mg kg$^{-1}$) | RMSE (% or mg kg$^{-1}$) | NRMSE (%) |
| Soil water content | MR0% | 0.36 | 0.41 | 2.13 |
| | MR1.5% | 0.48 | 0.55 | 2.56 |
| | MR3.0% | 0.53 | 0.59 | 2.53 |
| | MR4.5% | 0.71 | 0.93 | 3.74 |
| Nitrate–N | MR0% | 1.32 | 1.53 | 3.13 |
| | MR1.5% | 0.71 | 0.79 | 1.49 |
| | MR3.0% | 1.12 | 1.25 | 2.17 |
| | MR4.5% | 1.18 | 1.32 | 2.08 |
| Available P | MR0% | 0.08 | 0.1 | 0.95 |
| | MR1.5% | 0.09 | 0.12 | 0.91 |
| | MR3.0% | 0.09 | 0.11 | 0.67 |
| | MR4.5% | 0.84 | 0.99 | 5.68 |
| Available K | MR0% | 1.17 | 1.61 | 1.55 |
| | MR1.5% | 1.24 | 1.53 | 1.24 |
| | MR3.0% | 3.66 | 4.72 | 3.32 |
| | MR4.5% | 3.19 | 3.67 | 2.25 |

(1) MR means mixing ratios of waste biomass in soil; (2) *MAE*, *RMSE*, and *NRMSE* represent the root mean square error, mean relative error, and normalized root mean squared error of the fitted model, respectively.

## 4. Discussion

The infiltration of fertilizer solution and the distribution characteristics of water-fertilizer affect soil fertility and root growth of crops and ultimately affect water-fertilizer use efficiency under the fertigation technology. The study on infiltration and distribution of fertilizer solution in mixing waste biomass soil under moistube fertigation can provide a theoretical basis for the popularization and application of new moistube fertigation technology.

Soil water infiltration comprises a variety of complex processes that cannot be assessed by one parameter alone but requires multiple parameters [41,42]. In this study, the infiltration process of fertilizer solution was divided into three stages, and six infiltration parameters (IIR, AIRSI, AIRSII, AIRSIII, SIR, and AIR) were used to evaluate the infiltration rate of fertilizer solution in the soil with different waste biomass mixing ratios. The results showed that the mixing ratio had a significant effect on the infiltration rate and that mixing waste biomass in soil could significantly increase IIR, SIR, and AIR of moistube fertigation.

The cumulative infiltration of fertilizer solution for moistube fertigation is significantly increased by mixing waste biomass in soil, probably owing to that mixing waste biomass increased the organic matter content of soil and improved the size and distribution of soil pores, which reduced soil compaction, improved soil pore connectivity, and increased the infiltration gateway of fertilizer solution. This reduced the resistance of fertilizer solution flowing between the pore, thus enhancing cumulative infiltration [36,43]. Obviously, mixing waste biomass in soil could effectively improve the infiltration capacity of fertilizer solution in moistube fertigation.

Kostiakov infiltration model is commonly used to describe the infiltration process in farmland irrigation and the conservation of soil and water because of the great goodness of fit between the model and actual infiltration process [44–48]. In this study, this model was used to fit the infiltration process of fertilizer solution for moistube fertigation under different waste biomass mixing ratios and established the modified Kostiakov infiltration model with higher accuracy for fertilizer solution.

This study showed that the maximum content of water-fertilizer occurred just next to the moistube and that the water-fertilizer content gradually decreased further from the source, which was consistent with the results of related studies [20,26,28,29]. The distribution of soil water and nitrate–N were

similar in that distribution range were larger in wetted soil, which was mainly related to the strong movement ability of nitrate–N in soil with water [49,50]. Simultaneously, the distribution of available P and K tended to be the same and the distribution range was smaller, which was probably related to the strong adsorption of soil to available P and K [49,50]. It was also found that the distribution area of water-fertilizer in wetted soil was increased by mixing waste biomass, indicating that mixing moderate waste biomass in soil could enhance the movement ability of water and nutrient to a certain extent.

Waste biomass mixing ratio had a significant effect on the average content of water-fertilizer in wetted soil of moistube fertigation, which was mainly related to the increase of cumulative infiltration of fertilizer solution by mixing waste biomass. Mixing waste biomass could improve the distribution uniformity coefficients of soil water and nitrate–N in wetted soil, perhaps due to mixing waste biomass improving soil structure, accelerating fertilizer solution infiltration, increasing cumulative infiltration [36], and promoting the movement of water and nitrate–N far away from moistube [49], resulting in more uniform distribution of soil water and nitrate–N. However, the distribution uniformity coefficients of available P and K were significantly reduced by mixing waste biomass, which was related to the increase of infiltration rate of fertilizer solution and the enhancement of soil adsorption of available P and available K [51].

Logistic model is a common model for fitting the S-shaped curve, which has been widely used in many fields [52–54]. Compared with the traditional logistic model, the 4-parameter log-logistic model adds a parameter (inflection point) to make its fitting effect better [55,56]. In this study, the 4-parameter log-logistic model was used to fit the relationship between the content of water-fertilizer and horizontal distance in the wetting pattern of moistube fertigation and had a good fitting result.

During intermittent fertigation (such as flood irrigation, sprinkler irrigation, and drip irrigation), the soil surface undergoes an alternate of the absorption and desorption to form a dense layer. With the increase of intermittent infiltration frequency, the soil structure tends to be compact, which reduces the infiltration capacity of fertilizer solution [57–59]. However, moistube fertigation achieves a continuous supply of water-fertilizer with a small flow rate; thus, the infiltration of fertilizer solution is relatively uniform. According to the condition of soil and the growth characteristics of crops at different stages, moistube fertigation can effectively control the supply quantity and proportion of water and nutrient and can give full play to the coupling effect of water-fertilizer, so as to adjust water by fertilizer, to promote fertilizer by water, and to coordinate the supply of water-fertilizer. Compared with traditional fertilization methods, moistube fertigation can fulfill appropriately the crop's demand for water and nutrients and can provide a relatively stable water and fertilizer status for root growth [60]. It reduces the contact area between fertilizer and soil and realizes the integrated management of field moisture and nutrients, thereby improving the comprehensive utilization rate of water and fertilizer [61,62].

Moistube fertigation and agricultural waste biomass returning field are especially suitable for economic forest-planting areas where the distance of planting is large, soil organic matter and nutrient are poor, and drought is continual. This is mainly due to moistube fertigation being based on micro-nano porous membrane to achieve simultaneously irrigation and fertilization, and it does incompletely rely on soil suction to achieve fertigation [19,20]. The water-fertilizer distribution of moistube fertigation in the soil can be controlled by the pressure water head, moistube laying length, amended soil, and so on [24,25]. On the other hand, agricultural byproduct returning field can increase soil organic matter, can solve the problem of agricultural waste biomass, and then can realize sustainable development of agriculture. Thus, the technology will be more feasible in field application. Furthermore, the moistube fertigation system is a relatively simple structure, has low energy consumption, conveniently lays, and so on. It also can reduce the investment operation cost compared with drip and sprinkler fertilization systems [62]. Therefore, coupling moistube fertigation and agricultural waste biomass returning field has wide application prospects in agricultural resources cyclic utilization in water resource scarce regions.

## 5. Conclusions

Mixing waste biomass in soil increased significantly the infiltration rate and cumulative infiltration of fertilizer solution and improved the infiltration capacity of fertilizer solution under moistube fertigation. The relationship between the cumulative infiltration of fertilizer solution and infiltration time conformed to the Kostiakov infiltration model under different waste biomass mixing ratios.

Under moistube fertigation, the contour of water-fertilizer content was approximately ellipsoid around the moistube in wetted soil. The maximum content of water-fertilizer occurred just next to the moistube, and the content of water-fertilizer gradually decreased with the increase of horizontal distance. Mixing waste biomass in soil increased the distribution area and content of water-fertilizer in wetted soil.

Mixing waste biomass in soil effectively improved the distribution uniformity coefficient of soil water and nitrate–N but reduced that of available P and K in wetted soil. The 4-parameter log-logistic model fitted well with the distribution of soil water, nitrate–N, and available P and K in wetted soil of mixing waste biomass soil under moistube fertigation.

**Author Contributions:** Writing original draft preparation and editing full text, G.S.; Software and processing obtained data, Y.L.; Methodology and funding acquisition, X.G.; Writing-review, N.B., and Q.L.; Modify language structure, Y.L.

**Funding:** The present study was funded by National Natural Science Foundation of China (51979133, 51769010, 51469010, 51109102, and 51922072).

**Conflicts of Interest:** The authors declare no conflict of interest.

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
