# Peer review of "Effect of Moistube Fertigation on Infiltration and Distribution of Water-Fertilizer in Mixing Waste Biomass Soil"

_sustainability, doi:10.3390/su11236757_

Round 1

Reviewer 1 Report

Overall: This is a well-written article on a study of amending soil with waste biomass of various proportions to achieve an improved infiltration and distribution of water-fertilizer during moistube fertigation. It is a topic recent relevant discussion, given the expansion of this technology to realize water conservation goals in agriculture. The methods appear to be sound, the results are promising, and the conclusions are warranted. I have a few editing suggestions to make the writing a bit clearer and easier to read.

Specifically:

Line 17: comma after “(MR1.5%, MR3.0% and MR4.5%)”

Line 20: “by a modified infiltration model”

Line 21: “increasing the mixing ratio”

Line 22: “in amended wetting soil”

Line 24: “the Kostiakov infiltration model”

Line 25: “increase in waste biomass mixing ratio”

Line 28: “could provide a theoretical basis”

Line 34: “with fertilization to fertilize through”

Line 37: “technologies are the most widely used [2-4], and there are”

Line 40: comma after “factors”

Line 41: comma after “time”

Line 46: “outside of the membrane and soil suction. It delivers”

Line 47: “directly into the crop root zone continuously and slowly [16-17], thus providing and effective carrier”

Line 54: “reduced gradually further from the source,”

Line 55-56: Suggest: “Currently, the infiltration characteristics of fertilizer solution and the distribution of water-fertilizer in wetting soils during moistube fertigation are not known.”

Line 61-63: “have shown that comminuted straw amendment rapidly improves soil structure and increases the steady infiltration rate and cumulative infiltration of soil, while mixing in long straw decreases the soil infiltration capacity [32, 33].”

Line 66-67: “soil fertility; however, how mixing waste biomass (peanut shell) in the soil influences the infiltration”

Line 73 “so as to provide a theoretical basis”

Line 81: comma after “loam”; “Table 1 shows”

Line 98: “was arranged in a vertical insertion mode. The top of the moistube”

Line 102: “respectively” not needed here

Line 104: “All data in this paper are reported as means.”

Line 110: “The experimental data were recorded”

Line 112: comma after “stopped”

Line 113: comma after “method”

Line 113-114: “with a distance between sampling points of 5 cm.”

Line 116: “is shown in Figure 2.”

Line 122: “using a UV spectrophotometer.”

Line 124: “using a flame photometer.”

Line 127: “(AIRSI); likewise, the”

Line 150: “The accuracy of the fitted model was evaluated”

Line 175: “Given are the mean values and SD.”

Line 197: I think you mean “Eq.” instead of “Ep.”

Line 235-236: “available K, respectively, in wetting soil of moistube fertigation under different peanut shell mixing ratios.”

Line 283-284: “available P, available K, respectively, in wetting soils of moistube fertigation under different peanut shell mixing ratios.”

Line 295: “provide a theoretical basis”

Line 305-308: “which reduced soil compaction, improved soil pore connectivity, and increased the infiltration gateway of the fertilizer solution. This reduced the resistance of fertilizer solution flowing between the pores, thus enhancing cumulative infiltration”

Line 317: “gradually decreased further from the source, which was consistent”

Line 319: “similar in that distribution ranges were larger”

Line 321: “the same in that distribution ranges were smaller”

Line 342: “During intermittent fertigation”

Line 343: “alternate of absorption and desorption”

Line 345-347: “However, moistube fertigation achieves a continuous supply of water-fertilizer with a small flow rate, thus the infiltration of fertilizer solution is relatively uniform.”

Author Response

The first reviewer has given many suggestions to the manuscript. As follows:

Line 17: comma after “(MR1.5%, MR3.0% and MR4.5%)”

Line 20: “by a modified infiltration model”

Line 21: “increasing the mixing ratio”

Line 22: “in amended wetting soil”

Line 24: “the Kostiakov infiltration model”

Line 25: “increase in waste biomass mixing ratio”

Line 28: “could provide a theoretical basis”

Line 34: “with fertilization to fertilize through”

Line 37: “technologies are the most widely used [2-4], and there are”

Line 40: comma after “factors”

Line 41: comma after “time”

Line 46: “outside of the membrane and soil suction. It delivers”

Line 47: “directly into the crop root zone continuously and slowly [16-17], thus providing and effective carrier”

Line 54: “reduced gradually further from the source,”

Line 55-56: Suggest: “Currently, the infiltration characteristics of fertilizer solution and the distribution of water-fertilizer in wetting soils during moistube fertigation are not known.”

Line 61-63: “have shown that comminuted straw amendment rapidly improves soil structure and increases the steady infiltration rate and cumulative infiltration of soil, while mixing in long straw decreases the soil infiltration capacity [32, 33].”

Line 66-67: “soil fertility; however, how mixing waste biomass (peanut shell) in the soil influences the infiltration”

Line 73 “so as to provide a theoretical basis”

Line 81: comma after “loam”; “Table 1 shows”

Line 98: “was arranged in a vertical insertion mode. The top of the moistube”

Line 102: “respectively” not needed here

Line 104: “All data in this paper are reported as means.”

Line 110: “The experimental data were recorded”

Line 112: comma after “stopped”

Line 113: comma after “method”

Line 113-114: “with a distance between sampling points of 5 cm.”

Line 116: “is shown in Figure 2.”

Line 122: “using a UV spectrophotometer.”

Line 124: “using a flame photometer.”

Line 127: “(AIRSI); likewise, the”

Line 150: “The accuracy of the fitted model was evaluated”

Line 175: “Given are the mean values and SD.”

Line 197: I think you mean “Eq.” instead of “Ep.”

Line 235-236: “available K, respectively, in wetting soil of moistube fertigation under different peanut shell mixing ratios.”

Line 283-284: “available P, available K, respectively, in wetting soils of moistube fertigation under different peanut shell mixing ratios.”

Line 295: “provide a theoretical basis”

Line 305-308: “which reduced soil compaction, improved soil pore connectivity, and increased the infiltration gateway of the fertilizer solution. This reduced the resistance of fertilizer solution flowing between the pores, thus enhancing cumulative infiltration”

Line 317: “gradually decreased further from the source, which was consistent”

Line 319: “similar in that distribution ranges were larger”

Line 321: “the same in that distribution ranges were smaller”

Line 342: “During intermittent fertigation”

Line 343: “alternate of absorption and desorption”

Line 345-347: “However, moistube fertigation achieves a continuous supply of water-fertilizer with a small flow rate, thus the infiltration of fertilizer solution is relatively uniform.”

Response: We have revised according to the all suggestions.

Reviewer 2 Report

The submitted study focuses on the new technique of fertigation (fertilization through irrigation system) and investigates the process effectiveness through a series of indoor soil box experiments. The experimental setup allowed to investigate six infiltration parameters to evaluate the infiltration rate of fertilizer solution in the soil. Moreover, the effect of waste biomass (peanut shells) addition in three different mixing ratios was studied as well. The submitted manuscript is well written and contains a well-performed and described a statistical part. It could indeed provide a theoretical basis and reference for popularization and application of the new fertigation technology, however, considering limited scale of the experimental part (soil boxes of 40×40×45 cm) it would be definitely better suited for journals like Soil & Tillage Research or Irrigation Science.

To address challenges related to sustainability, e.g. sustainable utilization of resources such as land and water, the study should include also more general insight into proposed technique. For example, answering some of the following questions would make this manuscript more relevant to the Sustainability journal scope:

i)   What are the pros and cons of the fertigation technique? Especially when compared with regular granular applications

ii)Where this technique can be applied – it feasible when the variability between fields in the farm is high?

iii) What are the costs of the initial equipment of the fertigation installation in the field scale?

iv) How the results of your study can be transferred into the field application?

Author Response

Review 2

i) What are the pros and cons of the fertigation technique? Especially when compared with regular granular applications

Response: The fertigation technique has the characteristics of fast fertilizer efficiency, saving water and fertilizer, and reducing production cost, but relatively high the operating cost when compared with regular granular applications.

ii) Where this technique can be applied – it feasible when the variability between fields in the farm is high?

Response: Moistube irrigation is a new subsurface continuous irrigation technique, and addition of waste biomass is beneficial to improve soil structure. It was found that addition biomass in soil could improve infiltration efficiency and distribution of water and fertilizer. Therefore, it is feasible when the variability between fields in the farm is high.

iii) What are the costs of the initial equipment of the fertigation installation in the field scale?

Response: At present, our research is in the stage of indoor exploration, which has not yet been extended to the field. For this reason, we are unable to evaluate the cost of the equipment in the field scale.

iv) How the results of your study can be transferred into the field application?

Response: The present study found that addition waste biomass to the soil is beneficial to the improvement of the infiltration efficiency of the moistube fertigation and the uniformity of the soil nutrient distribution. On the one hand, agricultural byproduct returning field can solve the problem of agricultural waste biomass, and realize the sustainable development of agriculture. On the other hand, the moistube fertigation can meet appropriately the crop’s demand for water and nutrients. This irrigation method can improve the water and fertilizer use efficiency as well as to achieve the maximization of the economic benefit in the agricultural production. For this reason, the combination of the moistube fertigation with agricultural waste biomass returning technology has a wide application prospects in agriculture sustainability.

Reviewer 3 Report

The manuscript sustainability-618180, entitled “Effect of moistube fertigation on infiltration and distribution of water-fertilizer in mixing waste biomass soil” by Sun et al., deals with an interesting subject regarding the fertigation with moistube system in a soil were organic biomass was added with different ratios evaluating the nutrient spatial distribution. Moreover, the authors parametrize two different models useful to describe the water and nutrient infiltration processes assessing, also, their performance. The objectives of the study are interesting because the effect of this promising irrigation system on amended soil was not very thorough.

Although the manuscript is of good quality it needs some improvements to be published. First of all, I suggest the authors to carefully read and revise the language of the manuscript. Moreover, I suggest them to specify in the manuscript which marks are registered in accordance with the editorial rules of the magazine.

The introduction its good enough though in some passages too generic. The objective of the study clearly presented. Abstract clearly present the activities and the principal results.

Keywords are appropriate although I suggest to change or remove some of them.

Material and methods should be improved adding some references and some experimental aspects should be clarified.

Results are sufficiently detailed.

Discussion although is clear and comments the salient results obtained from the experiment it lacks of a clear explanation of what was observed and there is no comparison with other results from available bibliography.

Conclusions are clear, based on the results and concise.

Below are my specific comments:

In all manuscript: change “wetting soil” with “wetted soil” and when you specify the level of significance add a space after “P” (ex.: P <0.05).

Lines 15-19: Please rewrite. In this form, it is not very smooth. You could write as: "The infiltration rate and cumulative......were studied in three waste biomass mixing ratios taking a not amended soil as control”.

Line 25: change ”while that of” with "whereas".

Line 27: change “in wetting soil of mixing waste biomass soil” with "in mixing waste biomass wetted soil".

Line 30: delete “mixing ratio”; change “infiltration characteristics” with “nutrient infiltration”.

Lines 53-54: Please, rewrite or delete.

Lines 62-63: “the long straw” Wich kind od agricultural byproduct do you mean?

Line 77: change “tillage layer” with “superficial soil layer”.

Line 79: delete “naturally”; change “sieved (in a 2 mm sieve” with “sieved in order to pass through a 2 mm sieve”.

Line 81: add a reference about soil classification.

Line 83: delete “After air-drying, crushing and sieving” and rewrite as: “Sieved soil was mixed...”.

Line 83: Can you give some information about the peanut shell? mean dimension of particles, C/N ratio, level of degradation?

Line 85: delete spaces in chemical compounds.

Table 1: Change “Soil chemical composition” with “Soil texture (USDA)”; in “Soil initial nutrient” delete “initial”.

Line 104: Change “all data in this paper were average” with “Data presented are mean of 3 replicates.”.

Lines 105-108: How can you exclude that there has been a vertical movement of water due to the gravity?

Line 121: usually the 2 molar solution is used. Please check.

Lines 121-124: Please add some reference for the analytical methods applied.

Line 125: Change “Calculation formula” with “Calculation” or “Models applied”.

Line 152: replace the colon with commas.

Lines 154-157: Please delete.

Line 169: delete “The”.

Line 170: Add the measure unit inside brackets.

Line 175: Change “Give” with “Data”.

Line 176: Change “and SD” with “±SD”; “indicate statistically significant differences”.

Lines 181-184: Please move in the discussion section or delete.

Line 186: delete “The”.

Lines 192-194: Please move in the discussion section or delete.

Line 216: Change “by 24.0-78.4” with “from 24.0 to 78.4%.”.

Line 226-229: Move in the discussion section or delete.

Line 247: Change “higher” with “high”.

Line 248: Change “lower” with “low”.

Lines 252-254: delete.

Line 255: Delete “The”.

Table 4: Please check the formatting of this table. One decimal is sufficient.

Line 275: Change “smaller” with “small”.

Line 283: Change “wetting soil of moisture” with “wetted soil by moisture”.

Line 284: Delete “, respectively”; “MR means the mixing ratio….”.

Line 303: Change “could be” with “is”.

Line 305: Change “plant biomass” with "organic matter content".

Line 345: delete “and stable”.

Author Response

Review 3

In all manuscript: change “wetting soil” with “wetted soil” and when you specify the level of significance add a space after “P” (ex.: P <0.05).

Response: Yes, we have revised.

Lines 15-19: Please rewrite. In this form, it is not very smooth. You could write as: "The infiltration rate and cumulative......were studied in three waste biomass mixing ratios taking a not amended soil as control”.

Response: Yes, we have rewritten.

Line 25: change ”while that of” with "whereas".

Response: Yes, we have revised.

Line 27: change “in wetting soil of mixing waste biomass soil” with "in mixing waste biomass wetted soil".

Response: Yes, we have revised.

Line 30: delete “mixing ratio”; change “infiltration characteristics” with “nutrient infiltration”.

Response: Yes, we have deleted or revised.

Lines 53-54: Please, rewrite or delete.

Response: Yes, we have rewritten.

Lines 62-63: “the long straw” Wich kind old agricultural byproduct do you mean?

Response: It is wheat straw.

Line 77: change “tillage layer” with “superficial soil layer”.

Response: Yes, we have revised.

Line 79: delete “naturally”; change “sieved (in a 2 mm sieve” with “sieved in order to pass through a 2 mm sieve”.

Response: Yes, we have deleted or revised

Line 81: add a reference about soil classification.

Response: Yes, we have added

Line 83: delete “After air-drying, crushing and sieving” and rewrite as: “Sieved soil was mixed...”.

Response: Yes, we have deleted or rewritten

Line 83: Can you give some information about the peanut shell? mean dimension of particles, C/N ratio, level of degradation?

Response: The particle size of peanut shell is less than 2 mm. The fresh shell of peanuts were dried and triturated to mix in soil. The C/N ratio was not measured.

Line 85: delete spaces in chemical compounds.

Table 1: Change “Soil chemical composition” with “Soil texture (USDA)”; in “Soil initial nutrient” delete “initial”.

Response: Yes, we have deleted or revised.

Line 104: Change “all data in this paper were average” with “Data presented are mean of 3 replicates.”.

Response: Yes, we have revised.

Lines 105-108: How can you exclude that there has been a vertical movement of water due to the gravity?

Response: We did not exclude that there has been a vertical movement of water due to the gravity.

Line 121: usually the 2 molar solution is used. Please check.

Response: Yes, we have checked. The 1 molar solution is used in reference 5.

Lines 121-124: Please add some reference for the analytical methods applied.

Response: Yes, we have added.

Line 125: Change “Calculation formula” with “Calculation” or “Models applied”.

Response: Yes, we have revised.

Line 152: replace the colon with commas.

Response: Yes, we have revised.

Lines 154-157: Please delete.

Response: Yes, we have deleted.

Line 169: delete “The”.

Response: Yes, we have deleted.

Line 170: Add the measure unit inside brackets.

Response: Yes, we have revised.

Line 175: Change “Give” with “Data”.

Response: Yes, we have revised.

Line 176: Change “and SD” with “±SD”; “indicate statistically significant differences”.

Response: Yes, we have revised.

Lines 181-184: Please move in the discussion section or delete.

Response: Yes, we have deleted.

Line 186: delete “The”.

Response: Yes, we have deleted.

Lines 192-194: Please move in the discussion section or delete.

Response: Yes, we have deleted.

Line 216: Change “by 24.0-78.4” with “from 24.0 to 78.4%.”.

Response: Yes, we have revised.

Line 226-229: Move in the discussion section or delete.

Response: Yes, we have deleted.

Line 247: Change “higher” with “high”.

Response: Yes, we have revised.

Line 248: Change “lower” with “low”.

Response: Yes, we have revised.

Lines 252-254: delete.

Response: Yes, we have deleted.

Line 255: Delete “The”.

Table 4: Please check the formatting of this table. One decimal is sufficient.

Response: Yes, we have deleted and checked.

Line 275: Change “smaller” with “small”.

Response: Yes, we have revised.

Line 283: Change “wetting soil of moisture” with “wetted soil by moisture”.

Response: Yes, we have revised.

Line 284: Delete “, respectively”; “MR means the mixing ratio….”.

Response: Yes, we have deleted.

Line 303: Change “could be” with “is”.

Response: Yes, we have revised.

Line 305: Change “plant biomass” with "organic matter content".

Response: Yes, we have revised.

Line 345: delete “and stable”.

Response: Yes, we have deleted.

Round 2

Reviewer 2 Report

I am sorry to notice, but it seems like the Authors misunderstood comments included in my original review. The questions I've asked were supposed to address sustainability issues in the submitted manuscript, which in my opinion are simply missing. Therefore, the answers should be incorporated into the text, not just provided as a response to the reviewer comments.Since, none of the them have been addressed in the text I ask the Authors to reconsider my comments.

"To address challenges related to sustainability, e.g. sustainable utilization of resources such as land and water, the study should include also more general insight into proposed technique. For example, answering some of the following questions would make this manuscript more relevant to the Sustainability journal scope: i)   What are the pros and cons of the fertigation technique? Especially when compared with regular granular applications ii)Where this technique can be applied – it feasible when the variability between fields in the farm is high? iii) What are the costs of the initial equipment of the fertigation installation in the field scale?"

Author Response

1. I am sorry to notice, but it seems like the Authors misunderstood comments included in my original review. The questions I've asked were supposed to address sustainability issues in the submitted manuscript, which in my opinion are simply missing. Therefore, the answers should be incorporated into the text, not just provided as a response to the reviewer comments. Since, none of the them have been addressed in the text I ask the Authors to reconsider my comments.

Response: Yes, We have done them in introduction, such as the following: The average annual total of agricultural irrigation water is very high in China, but the water use efficiency is relative low. Similarly, China’s fertilizer use is high per years, but the utilization rate of fertilizer is low in the current season. Moreover, the excessive use of fertilizer increases the agricultural cost and causes environmental pollution. Research reported that there are many questions about cultivated land in China. For example, the area of inferior cultivated land was high, and the soil organic matter was decreased. Thus, it is great significant to employ both effective fertigation method and agricultural waste biomass returning field technology in China for improving the utilization efficiency of agricultural resources and achieving the sustainable development of agriculture.

2. What are the pros and cons of the fertigation technique? Especially when compared with regular granular applications

Response: Yes, We have supplemented the pros and cons of the fertigation technique, such as the following: Compared with traditional fertilization methods, moistube fertigation can fulfill appropriately the crop’s demand for water and nutrients, and provide a relatively stable water and fertilizer status for root growth. It reduces the contact area between fertilizer and soil and realizes the integrated management of field moisture and nutrients, thereby improving the comprehensive utilization rate of water and fertilizer. Regrettably, it is relatively high in the operating cost.

3. Where this technique can be applied – it feasible when the variability between fields in the farm is high?

Response: Yes, We have discussed them, such as the following: Moistube fertigation and agricultural waste biomass returning field are especially suitable for economic forest-planting areas where the distance of planting is large, soil organic matter and nutrient is poor, and drought is continual. This is mainly due to moistube fertigation is based on micro-nano semi-permeable membrane technology to achieve simultaneously irrigation and fertilization, and it does incomplete rely on soil suction to achieve fertigation. The water-fertilizer distribution of moistube fertigation in the soil can be controlled by the pressure water head, moistube laying length, amended soil and so on. On the other hand, agricultural byproduct returning field can increase soil organic matter and solve the problem of agricultural waste biomass, and then realize sustainable development of agriculture. Thus, the technology will be more feasible in field application.

4. What are the costs of the initial equipment of the fertigation installation in the field scale?

Response: Yes, We have discussed them, such as the following: The moistube fertigation system is relatively simple structure, low energy consumption, convenient laying and so on. It also can reduce the investment operation cost compared with drip and sprinkler fertilization systems

5. How the results of your study can be transferred into the field application?

Response: Yes, We have discussed them, such as the following: The present study found that adding waste biomass to the soil is beneficial to the improvement of the infiltration capacity of moistube fertigation and the uniformity of the soil nutrient distribution. On the one hand, agricultural byproduct returning field can increase soil organic matter and solve the problem of agricultural waste biomass, and then realize sustainable development of agriculture. On the other hand, the moistube fertigation can meet appropriately the crop’s demand for water and nutrients. This irrigation method can improve the water and fertilizer use efficiency as well as to achieve the maximization of the agricultural economic benefit. For this reason, coupling moistube fertigation and agricultural waste biomass returning field has a wide application prospects in agricultural resources cyclic utilization in water resource scarce regions.

Round 3

Reviewer 2 Report

The manuscript has been greatly improved after all changes made according to the reviewers comments. 

Author Response

We would like to thank you for your patience and time. Your comments have helped us a lot in revising this manuscript.